# Altered Fractional Amplitude of Low-Frequency Fluctuation in Anxious Parkinson’s Disease

**DOI:** 10.3390/brainsci13010087

**Published:** 2023-01-02

**Authors:** Peiyao Zhang, Yunpeng Gao, Yingying Hu, Yuan Luo, Lu Wang, Kang Wang, Hong Tian, Miao Jin

**Affiliations:** 1Department of Radiology, China-Japan Friendship Hospital, Beijing 100029, China; 2Department of Neurology, China-Japan Friendship Hospital, Beijing 100029, China; 3Department of Neurosurgery, China-Japan Friendship Hospital, Beijing 100029, China

**Keywords:** parkinson’s disease, anxiety, resting-state fMRI, fractional amplitude of low-frequency fluctuation

## Abstract

Objective: Anxiety symptoms are persistent in Parkinson’s disease (PD), but the underlying neural substrates are still unclear. In the current study, we aimed to explore the underlying neural mechanisms in PD patients with anxiety symptoms. Methods: 42 PD-A patients, 41 PD patients without anxiety symptoms (PD-NA), and 40 healthy controls (HCs) were recruited in the present study. All the subjects performed 3.0T fMRI scans. The fractional amplitude of low-frequency fluctuation (fALFF) analysis was used to investigate the alterations in neural activity among the three groups. A Pearson correlation analysis was performed between the altered fALFF value of the PD-A group and anxiety scores. Results: Compared with HCs, PD-A patients had higher fALFF values in the left cerebellum, cerebellum posterior lobe, bilateral temporal cortex, and brainstem and lower fALFF values in the bilateral inferior gyrus, bilateral basal ganglia areas, and left inferior parietal lobule. Moreover, between the two PD groups, PD-A patients showed higher fALFF values in the right precuneus and lower fALFF values in the bilateral inferior gyrus, bilateral basal ganglia areas, left inferior parietal lobule, and left occipital lobe. Furthermore, Pearson’s correlation analysis demonstrated that the right precuneus and left caudate were correlated with the Hamilton Anxiety Rating Scale scores. Conclusion: Our study found that anxiety symptoms in PD patients may be related to alterations of neurological activities in multiple brain regions. Furthermore, these may be critical radiological biomarkers for PD-A patients. Therefore, these findings can improve our understanding of the pathophysiological mechanisms underlying PD-A.

## 1. Introduction

Parkinson’s disease (PD) is a chronic neurodegenerative disease characterized by typical bradykinesia, resting tremors, rigidity, and loss of postural reflexes [1]. In addition to motor symptoms, PD is accompanied by non-motor symptoms such as cognitive impairment and psychiatric disorders [2], which may frequently precede motor symptoms [3]. Unfortunately, anxiety is one of the most common psychiatric disorders in PD patients and affects approximately 25–49% of PD patients [4]. Notably, anxiety is associated with increased severity of motor symptoms, which play a significant role in the aggravation of PD patients’ quality of life and daily function [5]. Previous neuronal circuit studies have established that dopaminergic uptake levels, norepinephrine transmission, and serotonergic neuron system dysfunction in the prefrontal cortex, striatum, cerebellum, and limbic system may be involved in the development of anxious symptoms in PD patients [6,7,8]. However, studies have not provided objective diagnostic markers to characterize the etiology of Parkinson’s disease with anxiety symptoms (PD-A), and whether PD and anxiety have common pathophysiological mechanisms is still poorly understood.

Recently, resting-state functional magnetic resonance imaging (rs-fMRI) has been applied as a non-invasive technique to explore the neuropathological basis of PD-A. Various analysis methods based on rs-fMRI were previously used in PD-A studies, in which specific changes were revealed in gray matter volume, regional or global neuronal spontaneous activity, and brain functional connectivity [5,9,10,11]. However, these conclusions were controversial and inconsistent, possibly due to the heterogeneity of the sample size and patient inclusion criteria, differences in data analysis, statistical methods, and other variables.

The fractional amplitude of low-frequency fluctuation (fALFF) method is one of the analysis methods of rs-fMRI. It is defined as the proportion of spontaneous low-frequency fluctuations (0.01–0.1 Hz) to the whole brain signal [12]. Compared with the low-frequency fluctuation (ALFF) method, the interference of physiological noise and the influence of non-neuronal activity-related factors can be effectively reduced. Additionally, the results may be more sensitive and specific [13]. Therefore, we compared the fALFF among PD-A patients, PD without anxiety symptoms (PD-NA) patients, and healthy controls (HCs) to characterize radiological biomarkers for PD with anxiety symptoms.

## 2. Materials and Methods

### 2.1. Participants

This is a prospective study, in which 83 PD patients were recruited from the department of neurology at the China-Japan Friendship Hospital according to the Movement Disorder Society (MDS) Clinical Diagnostic Criteria for Parkinson’s disease [14] between March 2019 and June 2021. The modified Hoehn and Yahr (H and Y) stage of all the PD patients was 1–2.5. The anxiety level of PD patients was estimated using the Diagnostic and Statistical Manual of Mental Disorders-IV (DSM-IV) criteria and separated into groups for PD-A and PD-NA based on a cut-off score of 7 on the Hamilton Anxiety Rating Scale (HAMA) score [15]. Additionally, all patients had first-onset anxiety symptoms and were not treated with medication. Meanwhile, 40 healthy controls (HCs) without psychological and neurological abnormalities or neuroimaging disturbances were enrolled in our research voluntarily. Participants were excluded if they had any evidence of cerebrovascular diseases, cognitive and mental disorders, serious respiratory, digestive, and cardiovascular diseases, and MRI contraindications. All scale scores were assessed through physician interviews of subjects and patient caregivers. All subjects were right-handed and voluntarily joined this study with informed consent. This study was approved by the ethics committee of the China-Japan Friendship Hospital.

### 2.2. Imaging Data Acquisition

MR images were acquired using a GE Discovery MR 750 3T scanner with an eight-channel head coil. All PD patients discontinued anti-Parkinsonian drugs 48 hours before the scan. Furthermore, all subjects were required to keep calm, relax, and stay awake with their eyes closed during the scanning. The RS-fMRI data were obtained using gradient echo-planar imaging (EPI) set to the following parameters: repetition time (TR)/echo time (TE) = 2500 ms/30 ms, flip angle = 90°, slice thickness/gap = 3.5 mm/0, slices = 42, acquisition matrix = 64 × 64, and FOV = 256 mm × 256 mm. Moreover, high-resolution whole-brain structural T1-weighted images were acquired with the parameters as follows: TR/TE = 6.7 ms/2.5 ms, flip angle = 12°, slice thickness/gap = 1 mm/1 mm, slices = 192, and field of view (FOV) = 256 mm × 256 mm.

### 2.3. Imaging Data Preprocessing

SPM12 and Data Processing and Analysis for (Resting-State) Brain Imaging (DPABI, Ver. 5; http://rfmri.org/dpabi) were used to preprocess the rs-fMRI data based on the MATLAB 2018a platform. First, the initial 10 volumes were discarded to eliminate the influence of unstable signals. Second, slice timing and head motion correction were performed on the remaining images. One PD-A and two PD-NA patients with excessive head motion (above 3 mm or 3^0^ in any direction) were excluded. Subsequently, the functional images were normalized to the Montreal Neurological Institute (MNI) space with an isotropic voxel size of 3 mm × 3 mm × 3 mm. Moreover, a linear regression analysis was used to remove the covariates, including white matter, cerebrospinal fluid, global signal, and Friston’s twenty-four head motion parameters. Finally, a low-pass frequency filter (0.01–0.08 Hz) was applied to reduce high-frequency physiological noise.

### 2.4. fALFF Analysis

The participants’ fALFF values were obtained using the DPABI software package. The preprocessed time series of each voxel was transformed into the frequency domain based on a fast Fourier transformation. The square root of each frequency was then calculated in the power spectrum to obtain the average square root at a low-frequency range (0.01–0.08 Hz). Thus, fALFF was acquired by comparing the ratio of the power spectrum of low-frequency (0.01–0.08 Hz) to the entire frequency range.

### 2.5. Statistical Analysis

A chi-square test was used for the gender data for all subjects in the current study. One-way analysis of variance (ANOVA) was applied for age, mini-mental state examination (MMSE), Hamilton depression rating scale (HDRS), and HAMA scores among the three groups. A two-sample t-test was performed to compare the two patient groups’ illness duration, modified H-Y stage, and UPDRS scores. The significance threshold was set to *p* < 0.05. Notably, all the above was analyzed using SPSS 22 statistical software (IBM Corp., Armonk, NY, USA).

An analysis of covariance (ANCOVA) was performed on the fALFF maps to identify brain areas with significant differences among the three groups. Post-hoc tests were used to compare the differences in fALFF within each pair of the three groups with the age, gender, disease course, H and Y stage, and MDS-UPDRS-III, MMSE, and HDRS scores were selected as covariates [voxel level *p* < 0.01, cluster level *p* < 0.05, determined by Gaussian random field (GRF) correction]. Furthermore, the clusters showing significant differences in fALFF between PD-A and PD-NA groups were extracted, and the mean fALFF *z*-values of these clusters were correlated with HAMA scores by Pearson correlation analysis. *p* < 0.05 was considered to indicate a statistically significant difference. All of these were performed by DPABI software.

## 3. Results 

Demographics and clinical data for each group are presented in Table 1. No significant differences among the three groups were found regarding age, gender, HDRS and MMSE scores (*p* > 0.05). Additionally, no significant differences were noted between the PD-A and PD-NA groups regarding disease duration, the UPDRS-III score, and modified H-Y scores (*p* > 0.05). Notably, a statistically significant difference among the three groups was observed in the HAMA score (*p* < 0.05).

### fALFF Analysis

Significant differences in fALFF values among the three groups were mainly located in the left cerebellum, cerebellum posterior lobe, bilateral frontal lobes, bilateral temporal lobes, left occipital lobe, left inferior parietal lobule, right caudate, and right precuneus (Table 2; Figure 1).

The higher fALFF values within the left cerebellum, posterior cerebellum lobe, bilateral temporal lobes, and brainstem, and the lower fALFF values within the bilateral frontal lobes, right basal ganglia areas, and left inferior parietal lobule were identified in PD-A patients compared with HCs (Table 3, Figure 2). Compared with HCs, PD-NA patients showed increased fALFF values in the left cerebellum and cerebellum posterior lobe. In contrast, decreased fALFF values were noted in the right cuneus and left superior parietal lobe (Table 3, Figure 2). When comparing the two PD groups, increased fALFF values of the right precuneus were found in PD-A patients. In contrast, decreased fALFF values were demonstrated in the bilateral frontal lobes, bilateral basal ganglia areas, left inferior parietal lobule, and left occipital lobe (Table 3, Figure 2). The fALFF values in the right precuneus (*r* = 0.54, *p* < 0.001) were positively correlated with HAMA scores of PD-A patients (Figure 3). Furthermore, the fALFF values in the left caudate were negatively correlated with HAMA scores of PD-A patients (*r* = −0.60, *p* < 0.001) according to Pearson’s correlation analysis (Figure 4).

## 4. Discussion

This article mainly focuses on the anxiety status of PD. Therefore, the core disease we are concerned about is PD, with anxiety as its only concomitant symptom. However, anxiety without PD is another disease. Thus, it was not included in the current study. Additionally, as anxiety was only a concomitant symptom of PD in this study, it was not subdivided into subgroups. Likewise, since we enrolled all patients in the early PD phase, no patient experienced motor fluctuations or dyskinesias. Previous studies mainly focused on the ALFFs in PD patients with non-motor symptoms [2,11,16]. This study compared the fALFF values among PD-A patients, PD-NA patients, and HCs. Consequently, we found that the altered fALFF values were mainly located in the right precuneus, bilateral inferior frontal gyrus, bilateral caudate/putamen, and left inferior parietal lobule. Moreover, the altered fALFF values of several brain regions were associated with HAMA scores in PD-A patients. Therefore, these findings will help us understand the pathogenesis of PD with anxiety symptoms during the early stages.

We found that decreased fALFF values of the left middle/inferior occipital gyrus and left inferior parietal lobule were indicated in PD-A patients compared with PD-NA patients. The occipital lobe is associated with integrating visual information and conscious processing [17,18], while the inferior parietal lobule controls visual attention and working memory [19,20]. Previous studies frequently reported that dysfunction of the two brain regions was involved in the development of non-motor disorders in PD patients [21,22,23,24]. Thus, the reduced fALFF value of the left occipital lobe and left inferior parietal lobule in PD patients may contribute to the development of anxiety symptoms via impairment of the image processing ability and cognitive function at the conscious level.

Our study demonstrated that the fALFF value of the bilateral inferior frontal gyrus was decreased in PD-A patients compared with PD-NA patients and HCs. The prefrontal lobe is the core brain area related to multiple emotional functions, including decision-making, emotional regulation, and social cognition [25,26]. The inferior frontal gyrus is in the lower part of the frontal lobe, which plays a vital role in dealing with emotional distraction and inhibitory control of cognitive function [27,28]. Notably, Zhang et al. reported that the functional connectivity (FC) between the right inferior frontal gyrus and left amygdala was negatively correlated with anxiety levels in PD patients [24]. Additionally, a structural study found that the level of anxiety was associated with a reduced volume of the bilateral prefrontal cortex in PD patients [29]. A regional homogeneity (ReHo) study also demonstrated that the neuronal activities of the left frontal lobe were decreased in PD-A patients [10]. Another study proposed that the increased anxiety severity was associated with a decreased FC between the amygdala and dorsolateral prefrontal cortex [9]. Furthermore, the results of our study were in agreement with these previous findings, suggesting an association between decreased fALFF values of the bilateral inferior frontal gyrus and the occurrence of emotional and cognitive impairment, which may induce anxiety symptoms in PD patients.

Typically, the striatum is responsible for regulating motor coordination. However, a new pathophysiological hypothesis proposes that the striatum is also involved in regulating and processing various emotional functions [25]. Previous neurotransmitter studies have suggested that higher anxiety levels in PD are associated with decreased dopamine transporter (DAT) binding in the bilateral caudate, the left putamen, and reduced noradrenaline transporter (NAT) in the left caudate [30,31]. A structural network study also reported that the severity of anxiety in PD patients was associated with reduced structural connectivity of the left amygdala with the right caudate [29]. Additionally, Wang et al. suggested that PD-A patients showed reduced FC between the bilateral putamen with the right orbitofrontal gyrus, left orbitofrontal gyrus, right cerebellum, and right precuneus, negatively correlating with HAMA scores [5]. In the current study, the fALFF value in the bilateral caudate/putamen nuclei was decreased in PD-A patients compared with PD-NA patients and HCs. Furthermore, the fALFF value of the left caudate was negatively correlated with HAMA scores. Therefore, these findings suggest that aberrant fALFF values in the contralateral caudate nucleus may disrupt the function of emotional management in PD patients, resulting in anxiety symptoms.

The precuneus is usually thought to be involved in self-concentration of attention, self-emotional regulation, and situational memory extraction [20]. Wee et al. found that increased anxiety severity was significantly associated with reduced bilateral precuneus gray matter volumes based on a voxel-based morphometry (VBM) study [32]. Moreover, a reduced FC between the right putamen and right precuneus was indicated in the PD-A group compared with the PD-NA group [10]. However, inconsistent with previous studies, we demonstrated that the fALFF value of the right precuneus was increased in PD-A patients and was positively correlated with HAMA scores. The precuneus is an essential node in the human brain and is extensively connected with multiple brain regions, including the prefrontal cortex, thalamus, and striatum [33]. Moreso, it is the functional core of the default mode network (DMN), which also encompasses the posterior cingulate cortex, medial prefrontal cortex, inferior parietal lobule, and temporal cortex [20]. In the current study, the decreased fALFF value was located in the prefrontal cortex, basal ganglia area, and inferior parietal lobule, all connected with the precuneus. Consequently, the increased fALFF value in the precuneus of PD patients may be to compensate for the declining fALFF values in other brain regions, which may lead to emotional regulation imbalances and abnormalities in the integration of cognitive and emotional processing, thus inducing anxiety symptoms.

In recent years, cerebellar dysfunction, particularly posterior cerebellar lobe dysfunction, has been linked to motor symptoms in PD and non-motor function [34]. Previous research has reported that the dysfunction of the cerebellum and posterior cerebellar lobe in PD patients may contribute to the development of anxiety symptoms [5,9,35]. This study found higher fALFF values in the left cerebellum and cerebellum posterior lobe in PD-A and PD-NA patients compared to HCs. However, these differences were not indicated in the comparison between PD-A and PD-NA patients. Therefore, it is suggested that cerebellar dysfunction may not be associated with the occurrence of anxiety symptoms in PD patients during the early stages. In addition, we also found higher fALFF values in the bilateral temporal lobe and brainstem when comparing PD-A patients and HCs, who frequently engaged in anxiety symptoms [2,35,36]. Nevertheless, the differences were not found in other pairwise comparisons, which may be due to the mild to moderate degree of anxiety symptoms among the PD patients selected in our study. Therefore, the temporal lobe and brainstem alterations between PD-A and PD-NA might be inhibited.

Several limitations should be noted regarding this study. First, to exclude some confounding factors such as depression and dementia, the sample size is relatively small, and the number of subjects in the three groups is not the same. Second, to explore the primary neural mechanisms underlying PD-A, only early-stage PD patients with anxiety symptoms were selected in our study. Consequently, these patients mainly presented with mild-to-moderate anxiety. Third, as this was a cross-sectional study, alterations in neural activities caused by the progress of PD-A were not seen. Therefore, a longitudinal follow-up study can be performed to explore further the neural mechanisms underlying PD-A. 

## 5. Conclusions 

Our study demonstrated that anxiety symptoms in PD may be related to altered neural activities in several brain regions, including the bilateral inferior frontal gyrus, bilateral caudate/putamen, left inferior parietal lobule, left occipital lobe and right precuneus. Notably, these brain regions may be critical radiological biomarkers for PD with anxiety symptoms and may thus improve our understanding of the pathophysiological mechanisms underlying PD-A.

## Figures and Tables

**Figure 1 brainsci-13-00087-f001:**
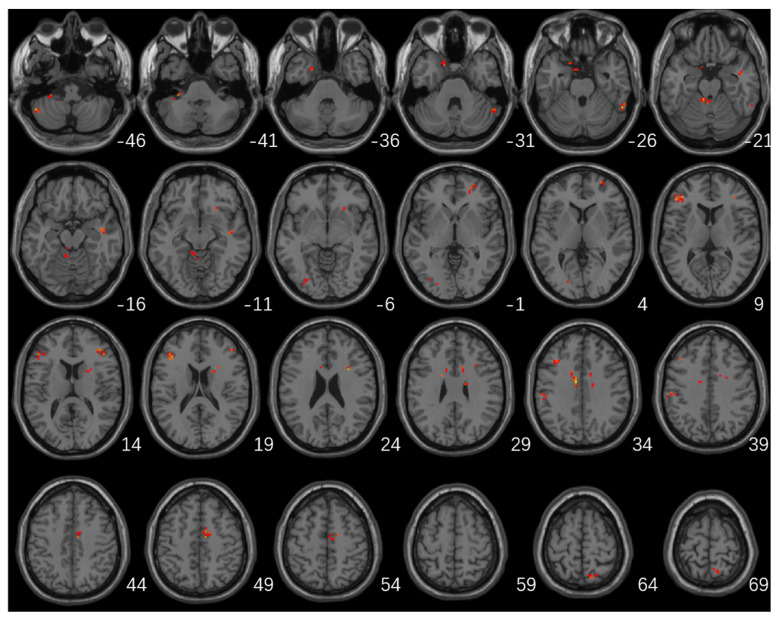
Brain regions with significant differences in fALFF among the three groups (GRF multiple comparison correction, *p* < 0.01 at the voxel level and *p* < 0.05 at the cluster level).

**Figure 2 brainsci-13-00087-f002:**
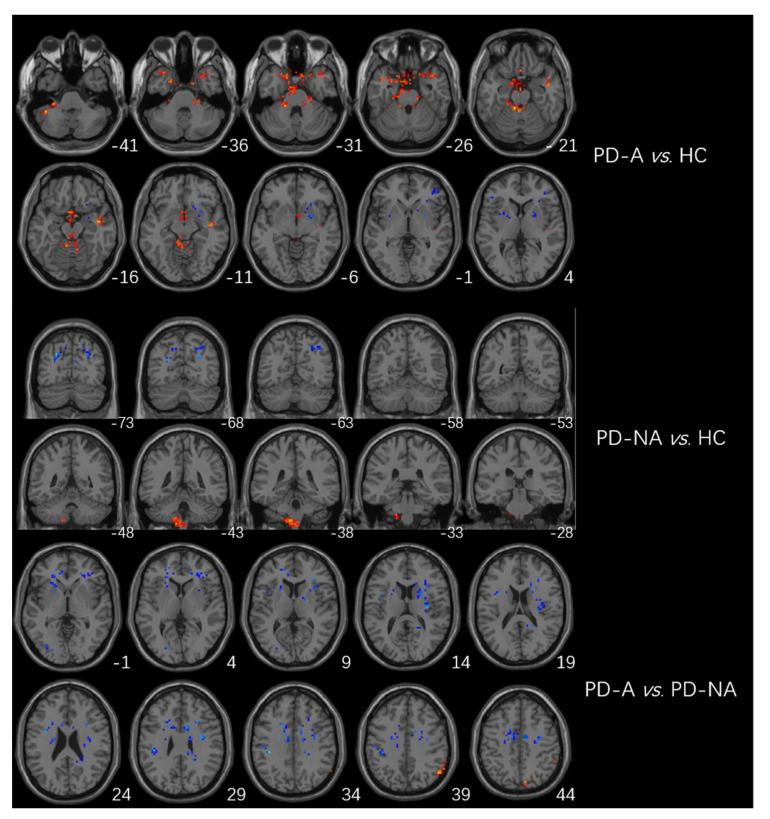
Brain regions showing differences in fALFF between groups. The red region represents the brain region with significantly increased fALFF (PD-A > HC, PD-A > PD-NA, and PD-NA > HC), and the blue region represents the brain region with significantly decreased DC (PD-A < HC, PD-A < PD-NA, and PD-NA < HC).

**Figure 3 brainsci-13-00087-f003:**
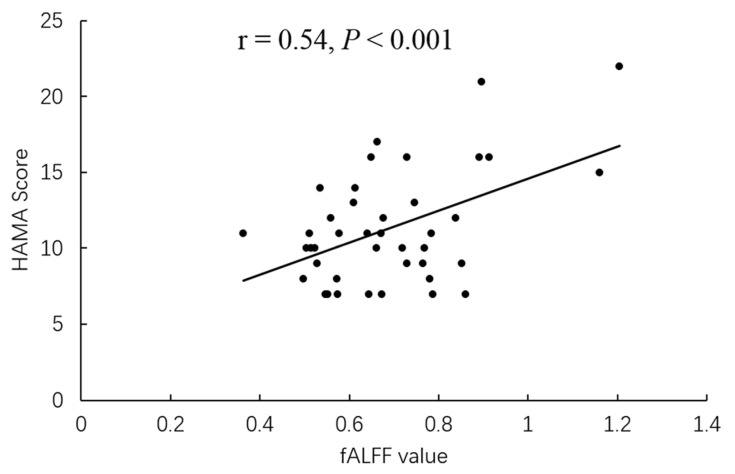
In PD-A patients, the fALFF value of the right precuneus showed a positive correlation with HAMA scores.

**Figure 4 brainsci-13-00087-f004:**
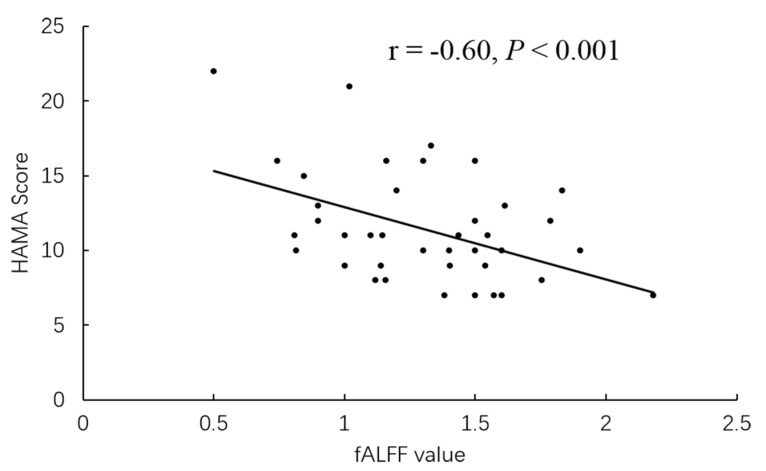
In PD-A patients, the fALFF value of the left caudate showed a negative correlation with HAMA scores.

**Table 1 brainsci-13-00087-t001:** Demographic and clinical characteristics of the samples.

Groups	PD-A (*n* = 41)	PD-NA (*n* = 39)	HC (*n* = 40)	*F/χ^2^/t*	*p*
Gender (M/F)	17/24	17/22	16/24	0.44	0.80 ^a^
Age (years)	65.54 ± 9.6	65.62 ± 10.23	60.97 ± 8.82	2.45	0.091 ^b^
Disease course (months)	39.31 ± 27.42	40.53 ± 35.37	-	−0.174	0.96 ^c^
MDS-UPDRS-score	28.45 ± 12.92	21.80 ± 12.96	-	0.75	2.32 ^c^
Hoehn and Yahr stage	1.85 ± 0.59	1.73 ± 0.55	-	1.05	0.91 ^c^
HAMA score	11.26 ± 3.70	3.12 ± 2.0	2.50 ± 1.30	12.27	0.005 ^b,^*
MMSE score	27.43 ± 1.58	28.22 ± 2.15	27.78 ± 1.66	1.99	0.14 ^b^
HAMD score	3.55 ± 1.88	3.00 ± 1.48	3.45 ± 1.69	1.31	0.27 ^b^

^a^*χ*^2^ test; ^b^ one-way ANOVA test; ^c^ two independent samples; *t* test; * *p* < 0.05.

**Table 2 brainsci-13-00087-t002:** Brain regions with significant differences in fALFF among the three groups.

Brain Regions	Coordinates of Maximum			Cluster Size	*F*
	X	Y	Z		
Left superior/middle temporal gyrus	21	3	−36	26	8.64
Left cerebellum/cerebellum posterior lobe	−30	−33	−42	23	10.38
Right superior/inferior temporal gyrus	57	−51	−27	23	13.00
Left middle/inferior occipital lobe	−33	−81	−3	13	8.33
Left inferior frontal gyrus	−39	30	18	29	12.44
Right inferior frontal gyrus	45	33	15	16	13.85
Right superior frontal gyrus	27	60	3	12	10.34
Right caudate	30	12	24	13	11.37
Left inferior parietal lobule	−54	−24	36	15	10.43
Left middle frontal gyrus	−39	24	36	12	12.07
Right precuneus	6	−63	66	13	9.62

**Table 3 brainsci-13-00087-t003:** Brain regions with significant between-group DC differences.

Brain Regions	Coordinates of Maximum			Cluster Size	*t*-Value
	X	Y	Z		
**PD-A > HC**					
Left cerebellum/cerebellum posterior lobe	−42	−45	−42	38	4.74
Right middle/superior temporal gyrus	42	−6	−15	71	4.29
Left middle/superior temporal gyrus	−36	17	−15	−37	4.02
Brainstem/midbrain	−9	−38	−21	48	4.48
**PD-A < HC**					
Right inferior frontal gyrus	39	39	15	24	−4.21
Left inferior frontal gyrus	−42	27	18	50	−4.19
Left putamen/caudate	−33	9	6	24	−3.94
Right putamen/caudate	18	12	−9	24	−4.23
Left inferior parietal lobule	−54	−24	36	22	−4.36
**PD-NA > HC**					
Left cerebellum/cerebellum posterior lobe	−12	−42	−60	21	4.29
**PD-NA < HC**					
Right cuneus	6	−78	21	40	−4.18
Left superior parietal lobe	−24	−72	33	18	−4.99
**PD-A > PD-NA**					
Right precuneus	6	−72	54	39	3.99
**PD-A < PD-NA**					
Left middle/inferior occipital gyrus	−36	−68	12	42	−4.91
Left inferior frontal gyrus	−27	27	−12	41	−3.76
Right inferior frontal gyrus	30	30	9	40	−3.95
Left putamen/caudate	−33	9	6	24	−3.94
Right caudate/putamen	39	12	6	24	−4.22
Left inferior parietal lobule	−39	−30	33	28	−4.53

## Data Availability

The data presented in this study are available upon request from the corresponding author.

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
