# Peer review of "Altered Fractional Amplitude of Low-Frequency Fluctuation in Anxious Parkinson’s Disease"

_brainsci, 2023, doi:10.3390/brainsci13010087_

Round 1

Reviewer 1 Report

The study is very interesting. However, its quality should be enhanced by adding a "control" group of patients with anxiety without PD. This should be useful to discriminate if the results found were related with the presence of anxiety. In addition, it should be useful to specify how many patients had "panic" or "anxiety" attacks (see reference PMID: 8424307) and describe this subgroup separately, and incicate how many patients showed motor fluctuations, dyskinesias and both. 

Author Response

The study is very interesting. However, its quality should be enhanced by adding a "control" group of patients with anxiety without PD. This should be useful to discriminate if the results found were related with the presence of anxiety. In addition, it should be useful to specify how many patients had "panic" or "anxiety" attacks (see reference PMID: 8424307) and describe this subgroup separately, and indicate how many patients showed motor fluctuations, dyskinesias and both. 

Response: Thank you for your valuable comments. In this article, we mainly focus on the anxiety status in Parkinson's disease. Therefore, the core disease we are concerned about is Parkinson's disease, and anxiety is only its concomitant symptom. However, anxiety without PD is another disease, thus it was not included in the current study. Moreover, as anxiety was only a concomitant symptom of PD in this study, it was not subdivided into subgroups. In addition, since we enrolled all patients in the early PD phase, no patient experienced motor fluctuation or dyskinesias.

Reviewer 2 Report

In the manuscript titled ‘Altered fractional amplitude of low-frequency fluctuation in anxious Parkinson’s disease’, Peiyao Zhang et al. has carefully analyzed and reported the changes in fMRI signals in the brain regions of patients with early Parkinson’s disease (PD) with anxiety. The authors have selected a particular normalized low intensity fMRI signal which was widely reported in number of neurological and neuropsychiatric conditions but not in the condition of interest of the current manuscript. This is indeed a sincere effort by the authors which led to some important findings related to the neural correlates of anxiety present in PD patients.

I am curious to know whether the diagnosis has been confirmed by the psychiatrist as Generalized Anxiety Disorders, as per DSM IV criteria. It is important to convey whether the patients were diagnosed GAD patients as per the criteria, or they just have symptoms of anxiety. It is also necessary to include appropriate reference supporting the cut-off selection in the HAMA score. How the questionnaire was administered is important to know (self-administered/ interviewed). Whether the care givers also assisted the filling up of the questionnaire.

The authors have mentioned that the patients were devoid of depression and dementia. To support that that they have mentioned mean MMSE score to characterize the cognitive profiles in all three groups. Similarly, the Depression scores should also be mentioned in Table 1.

Additionally, deep clinical characterization the patients are necessary. For example, the phenotype of PD, drug usage, comorbidities, UPDRS I, II, III, IV scores separately, presence of dyskinesia (UPDRS IV) are some examples which is recommended to include in the table 1. Details about other neuro-psychiatric symptoms are also essential. For example, presence of hallucination may influence the occipital fMRI signals significantly. Whether the anxiety is treated through some pharmacological/nonpharmacological means should also be taken into consideration during interpretation of the result. If any consistency is maintained regarding drug free interval [or clinical phase (on/off)] before fMRI data acquisition should also be mentioned.

Apart from these clinical details, it will be also interesting to know spatial distribution of the clusters, which was presented as the cluster size in the manuscript. Is it possible to mention the percent involvement too? For example, the total voxel of posterior cerebellum (right) is x and significant voxels are y. Hence the y/x*100 percent significant voxels may be reported.

I am unsure how the authors concluded that cerebellar impairment is associated with dyskinesia in the early stages of PD (line 253). The authors may clarify.

Overall, it is an interesting study. 

Author Response

I am curious to know whether the diagnosis has been confirmed by the psychiatrist as Generalized Anxiety Disorders, as per DSM IV criteria. It is important to convey whether the patients were diagnosed GAD patients as per the criteria, or they just have symptoms of anxiety. It is also necessary to include appropriate reference supporting the cut-off selection in the HAMA score. How the questionnaire was administered is important to know (self-administered/ interviewed). Whether the care givers also assisted the filling up of the questionnaire.

Response: Thanks for your comments. In this study, DSM-IV and Hamilton Anxiety Rating Scale (HAMA) was used to diagnose PD patients with concomitant anxiety symptoms. Unlike GAD, the anxiety symptoms in this study were only concomitant symptoms based on Parkinson's disease. The reference supporting the cut-off selection in the HAMA was cited in the manuscript. All scale scores were assessed through physician interviews with subjects and patient caregivers. The above content has been added in the revised manuscript.

The authors have mentioned that the patients were devoid of depression and dementia. To support that that they have mentioned mean MMSE score to characterize the cognitive profiles in all three groups. Similarly, the Depression scores should also be mentioned in Table 1.

Response: Thanks for your comments. Depression scores have been added in the table1.

Additionally, deep clinical characterization the patients are necessary. For example, the phenotype of PD, drug usage, comorbidities, UPDRS I, II, III, IV scores separately, presence of dyskinesia (UPDRS IV) are some examples which is recommended to include in the table 1. Details about other neuro-psychiatric symptoms are also essential. For example, presence of hallucination may influence the occipital fMRI signals significantly. Whether the anxiety is treated through some pharmacological/nonpharmacological means should also be taken into consideration during interpretation of the result. If any consistency is maintained regarding drug free interval [or clinical phase (on/off)] before fMRI data acquisition should also be mentioned.

Response: Thanks for your comments. In this article, we mainly focus on the fALFF alterations in Parkinson's disease patients with anxiety symptoms and discussed more related fALFF analysis. The deep clinical features of PD-A patients will be analyzed in detail in other articles in the future. In addition, the patients included in this study were the patients who were diagnosed as not taking medicine for the first time and the patients who took medicine regularly and stopped taking medicine for 48 hours before scanning. All patients did not have any neuropsychological abnormality other than anxiety symptom, and the anxiety symptoms occurred for the first time without treatment. The above content has been added in the revised manuscript.

Apart from these clinical details, it will be also interesting to know spatial distribution of the clusters, which was presented as the cluster size in the manuscript. Is it possible to mention the percent involvement too? For example, the total voxel of posterior cerebellum (right) is x and significant voxels are y. Hence the y/x*100 percent significant voxels may be reported.

Response: Thanks for your comments. The clusters are composed of voxels, and there is no percent involvement.

I am unsure how the authors concluded that cerebellar impairment is associated with dyskinesia in the early stages of PD (line 253). The authors may clarify.

Response: Thanks for your comments. We are very sorry that we expressed it incorrectly. What we want to express is that cerebellar dysfunction in early PD patients may not be related to the occurrence of anxiety symptoms, and these contents have been modified in the manuscript.

Round 2

Reviewer 1 Report

I suggest as minor change that the authors summarizes data commented in their author reponse in the discussion ("In this article, we mainly focus on the anxiety status in Parkinson's disease. Therefore, the core disease we are concerned about is Parkinson's disease, and anxiety is only its concomitant symptom. However, anxiety without PD is another disease, thus it was not included in the current study. Moreover, as anxiety was only a concomitant symptom of PD in this study, it was not subdivided into subgroups. In addition, since we enrolled all patients in the early PD phase, no patient experienced motor fluctuation or dyskinesias").

Author Response

Thanks for your comment. What you suggested was added to the document.
